# New Opportunities for Endometrial Health by Modifying Uterine Microbial Composition: Present or Future?

**DOI:** 10.3390/biom10040593

**Published:** 2020-04-11

**Authors:** Nerea M. Molina, Alberto Sola-Leyva, Maria Jose Saez-Lara, Julio Plaza-Diaz, Aleksandra Tubić-Pavlović, Barbara Romero, Ana Clavero, Juan Mozas-Moreno, Juan Fontes, Signe Altmäe

**Affiliations:** 1Department of Biochemistry and Molecular Biology, Faculty of Sciences, University of Granada, 18071 Granada, Spain; molinanerea@ugr.es (N.M.M.); albertosola@ugr.es (A.S.-L.); mjsaez@ugr.es (M.J.S.-L.); 2Instituto de Investigación Biosanitaria ibs.GRANADA, 18014 Granada, Spain; jrplaza@ugr.es (J.P.-D.); barbararg79@yahoo.es (B.R.); anaclaverogilabert@gmail.com (A.C.); jmozas@ugr.es (J.M.-M.); juanfontesjimenez@gmail.com (J.F.); 3“José Mataix Verdú” Institute of Nutrition and Food Technology (INYTA), Biomedical Research Centre (CIBM), University of Granada, 18100 Granada, Spain; 4Department of Biochemistry and Molecular Biology II, Faculty of Pharmacy, University of Granada, 18011 Granada, Spain; 5Clinic of Gynecology and Obstetrics, Clinical Center Niš, 18000 Niš, Serbia; ackana@medianis.net; 6Unidad de Reproducción, UGC de Obstetricia y Ginecología, Hospital Universitario Virgen de las Nieves, 18014 Granada, Spain; 7Consortium for Biomedical Research in Epidemiology & Public Health (CIBER Epidemiología y Salud Pública-CIBERESP), 28029 Madrid, Spain; 8Departament of Obstetrics and Gynecology, Faculty of Medicine, University of Granada, 18016 Granada, Spain; 9Competence Centre on Health Technologies, 50410 Tartu, Estonia

**Keywords:** antibiotics, chronic endometritis, endometriosis, endometrium, microbiome, microbiota, prebiotics, probiotics, uterus

## Abstract

Current knowledge suggests that the uterus harbours its own microbiota, where the microbes could influence the uterine functions in health and disease; however, the core uterine microbial composition and the host-microbial relationships remain to be fully elucidated. Different studies are indicating, based on next-generation sequencing techniques, that microbial dysbiosis could be associated with several gynaecological disorders, such as endometriosis, chronic endometritis, dysfunctional menstrual bleeding, endometrial cancer, and infertility. Treatments using antibiotics and probiotics and/or prebiotics for endometrial microbial dysbiosis are being applied. Nevertheless there is no unified protocol for assessing the endometrial dysbiosis and no optimal treatment protocol for the established dysbiosis. With this review we outline the microbes (mostly bacteria) identified in the endometrial microbiome studies, the current treatments offered for bacterial dysbiosis in the clinical setting, and the future possibilities such as pro- and prebiotics and microbial transplants for modifying uterine microbial composition.

## 1. Introduction

For a long time it was assumed that the uterus is a sterile organ, with microbial colonisation present only in infection or in a pathological process [1]. The ‘sterile womb’ hypothesis has been challenged by recent studies using next-generation sequencing (NGS) where unique uterine microbial composition was detected. As less than 1% of microbes can grow and form colonies on agar plates [2], analysis of the genomes of the microorganisms, i.e., microbiome, overcomes two common limitations of the traditional culture-based microbe characteristics: nonculturability and genomic diversity [3], opening a new research field in reproductive medicine. 

The initial studies of endometrial microbiome suggest its association with reproductive outcomes in assisted reproduction [4,5,6,7,8,9,10,11,12,13] and with different gynaecological pathologies such as chronic endometritis (CE) [14,15,16], endometriosis [13,17,18,19,20,21], dysfunctional endometrial bleeding [22], endometrial polyps [16], and endometrial cancer or hyperplasia [23,24,25]. Nevertheless, causality has been difficult to prove because the reproductive tract represents a polymicrobial niche [26], and it is not clear whether dysbiosis within the uterus is a cause or a consequence of a pathology. 

Furthermore, uterine microbial transmission is not clearly established, whereas different routes were proposed, including ascension of bacteria through the cervix, retrograde spread through fallopian tubes, haematogenous spread of oral and/or gut bacteria, through gynaecological procedures (e.g., assisted reproductive technology-related procedures; insertion/removal of the intrauterine devices), sexual habits, and/or with sperm [27,28,29,30,31]. Ascension of radioactively labelled macrospheres from vagina to the uterus by a ‘uterine peristaltic pump’ activity was shown [32,33], highlighting the most probable way for bacterial route into the uterus. Indeed, different studies suggest that microbial composition is highly influenced by the vaginal microbes [8,11,34,35,36].

Evidence for existence of certain microbiota (i.e., microbial community) in the uterus of healthy women is accumulating [14,37,38,39,40,41], but the confirmation of living microbes in the uterus is controversial and, up to now, no consensus on the core uterine microbial composition can be deduced. Uterine microbiome studies are challenged by many factors starting from the hormonal and physiological changes within the menstrual cycle, difficulty in obtaining uterine samples without contaminating the sample with vaginal/cervical bacteria, up to a high contamination risk also during samples processing and sequencing (see Figure 1 for factors). Uterine sample requires invasive sampling methods, and even when avoiding biopsy via the cervix using surgical or explorative procedures, women undergoing these procedures usually present existing medical conditions and often are peri- and postmenopausal, which could contribute to the compositional shifts in the microbiome [42]. Furthermore, the NGS-based studies conducted so far focused on detecting microbial DNA sequences. While this approach provides knowledge of possible taxa present and describes the microbiome, the mere presence of a DNA sequence does not equate with the presence of a live bacteria. Nevertheless, there are studies showing that short bacterial DNA fragments and microbial components can induce a physiological inflammatory response in the host [43,44,45]. Furthermore, other NGS methods are being applied in order to detect active microorganisms, such as metatranscriptomics which analyses microbial RNA transcripts, identifying thereby potential functionally active microorganisms [46,47]. Clearly more knowledge about the microbial composition, dynamics, and function within the uterus in health and disease is warranted in order to better understand the microbial homeostasis and dysbiosis in endometrial functions. 

In this review, we summarise the microorganisms identified in microbiome studies in human endometrium, the current treatments offered for bacterial dysbiosis in clinical setting, and the future possibilities for modifying uterine microbial composition.

## 2. Possible Function of Microbes in the Uterus 

It is becoming increasingly evident that microorganisms play an important role in our health and well-being through the production of bioactive molecules shaping a healthy microbiota, which in turn, interacts with our own cells to regulate and influence our metabolism, physiology, and immune functions that ultimately shape our health and resistance to a disease [94,95,96]. The exact role and relationship between microbes and the female reproductive tract remain to be established.

Endometrium seems to be an immunologically suited niche for microbiota with its possible function in modulating inflammatory and immune responses [30,37,97] (Figure 2). The mucosal layers of the female reproductive tract constitute a part of the mucosal immune system, which exhibits a broad repertoire of immune responses [98]. Further support for the possible immune responses in the uterus is provided by the fact that the genital epithelial cells express a wide range of pattern recognition receptors (PRRs) that promote their ability to recognise and differentially respond to various pathogens [98]. The PRRs found in the female reproductive tract include Toll-like receptors (TLRs) and NOD-like receptors, which have important roles in protecting against pathogenic invasion, in tissue adaptation and ultimately in successful reproduction [37,98]. Also pathophysiological effects of uterine microbes on the endometrial epithelium were proposed: (1) the genomic stability of uterine epithelia could be impacted via modulation of transcription factors and other genomic and epigenetic alterations; (2) the integrity of the epithelial barrier could be impaired; and (3) the microbial-secreted metabolites and the inflammation triggered by TLR activation could lead to suppression and/or overgrowth of specific bacteria [30]. The potential molecular functions of the endometrial microbes were linked to cell metabolism, motility, genetic information, immune system, and signalling processes in the NGS studies [5,20]. Regardless of all these hypotheses, future studies are needed to identify the basal uterine microbiota, its diversity and its functions and interactions with the endometrium.

## 3. Uterine/Endometrial Microbial Composition in Health and Disease

Several studies demonstrated that asymptomatic women harbour commensal microbial communities in their uterus [8,11,16,22,102], and that the uterine microbiome seems to be altered in women who suffer gynaecological pathologies such CE [14,15,16], endometriosis [17,19,20], dysfunctional endometrial bleeding [22], endometrial polyps [16]⁠, endometrial cancer or hyperplasia [16,23,24,25], and infertility [4,8,11,13]⁠ (Table 1). All these studies support the evidence that the uterine microbial composition is clinically relevant and requires further investigation.

### 3.1. Healthy Women

The uterine microbiome of ‘healthy’ women was mostly investigated as a control group in studies of infertile women or those with a gynaecological pathology [7,8,11,13,14,15,16,17,18,19,22,23,24] (Table 1). Nevertheless, these data suggest that commensal microorganisms can inhabit the upper reproductive tract of healthy women. Regardless of the growing body of literature, the core uterine microbial composition remains an open issue. There are studies concluding that the endometrium has a resident microbiome dominated by *Lactobacillus* species being similar to that of the vagina [8,11,16], while other studies from surgical procedures (where vaginal bacterial contamination is minimised) demonstrate that *Lactobacillus* is rare and that *Acinetobacter*, *Pseudomonas* and *Comamonadaceae* dominate [20,25,103]. Also a previous study analysing virgo intacta women identified the obligate anaerobes *Jonquetella* and *Fusobacterium* together with *Preovotella* as the predominant taxa in endometrium [22]. Furthermore, as 40% of the endometrial samples obtained by abdominal hysterectomy did not present any detectable microbes above the negative controls [25], this raises further doubts as to whether there is a unique uterine microbiome in all women, whether these detected bacteria are temporarily or permanently in the uterus, or whether we are dealing with contamination.

### 3.2. Infertility

Several publications support the theory that alterations in the endometrial microbiome may also impact the reproductive potential of infertile patients and perhaps, correcting microbial dysbiosis would lead to improve success [105]. There are numerous studies trying to associate endometrial microbiome with assisted reproduction outcomes, but just one has found a statistically significant difference in microbiome composition with success in infertility treatment outcomes [11]. In that study, *Lactobacillus* dominance (>90% of all bacteria) correlated positively with embryo implantation, pregnancy and live birth rates among infertile women undergoing *in vitro* fertilisation (IVF) [11]. Other studies have not detected any significant associations between endometrial microbiome and the treatment outcomes [4,6,7,8,9,10,12,13]. 

### 3.3. Endometriosis

The pathophysiology of endometriosis is still unclear [106]. The ‘Bacterial contamination’ hypothesis in endometriosis was proposed [107], where the inflammatory mediator lipopolysaccharide could be the initial trigger and bacterial contamination its source in the intrauterine environment serving as the primary cause in the growth regulation of endometriosis [108]. Also, considering the altered inflammatory condition in this disease, it is probable that the microbial pathogens activate the immune response by binding to host receptors. Indeed, complex bidirectional interaction between the microbiome and endometriosis is gaining evidence [109]. Nevertheless, studies on endometrial microbiome and endometriosis yielded contradicting results: *Streptococcaceae*, *Moraxellaceae*, *Staphylococaceae* and *Enterobacteriaceae* families were significantly increased while *Lactobacillus* species decreased in samples obtained from women with endometriosis [19], and uterine wash samples from women both with and without endometriosis detected *Lactobacillus* together with *Barnesiella*, *Flavobacterium* and *Pseudomonas* as the most predominant genera [18]. Another study investigating women with endometriosis identified the presence of *Pseudomonas, Acinetobacter, Vagococcus and Sphingobium* in uterus and revealed that uterine microbiota composition is significantly different in infertile women due to endometriosis [20]. A recently published study analysing endometriotic lesions found that the microbial diversity of lesions was higher compared to eutopic endometrium, where *Lactobacillus*, *Enterococcus*, *Gardnerella*, *Pseudomonas*, *Alishewanella*, *Ureaplasma* and *Aerococcus* prevailed [17].

### 3.4. Chronic Endometritis

CE is a medical condition implicated in 12–46% cases of infertile patients [110]. Indeed, different studies demonstrated that CE is favourably prevalent in patients affected by infertility, especially in case of repeated IVF failures [111]. In CE, the persistent inflammation of the endometrial mucosa is caused by the presence of bacterial pathogens in the uterine cavity [112]. The most common responsible bacteria for CE are *Enterococcus faecalis*, *Enterobacteriaceae*, *Streptococcus* spp., *Staphylococcus* spp., *Gardnerella vaginalis*, *Mycoplasma* spp. and other pathogens associated with sexually transmitted infections, such as *Ureaplasma urealyticum*, *Chlamydia trachomatis*, and *Neisseria gonorrhoeae* [14].

### 3.5. Endometrial Polyps

Endometrial polyps, a common gynaecologic disease featured as a localised overgrowth of mucosa, was correlated with CE, and continuous stimulation of biological inflammatory factors are believed to contribute to the disease [113,114]. A previous study found increased *Lactobacillus, Bifidobacterium*, *Gardnerella*, *Streptococcus*, *Alteromonas* and *Euryarchaeota* (Archaea) and decreased *Pseudomonas* and *Enterobacteriaceae* among endometrial samples from women with endometrial polyps when compared to non-diseased women [16] (Table 1). 

### 3.6. Dysfunctional Menstrual Bleeding

Distinct microbial communities are believed to have a role in gynaecological pathologies such as menorrhagia and dysmenorrhea [22]. The study identified *Lactobacillus* dominance together with *Gardnerella vaginalis*, *Veillonella* spp., *Prevotela* spp., and *Sneathia* spp. in the endometrial samples (Table 1), and also detected menstrual cycle dependent changes within the endometrial microbiome during the proliferative and secretory phases of the cycle [22].

### 3.7. Endometrial Cancer

Prevailing hypotheses for endometrial cancer aetiology are focused on obesity and hormones, and do not consider the potential role of endometrial microbiota [24]. However, studies in other fields, such as cervical cancer, prostate cancer and gastrointestinal tumours, demonstrated that microorganisms play important roles in cancer causation and development [31,115,116,117,118]. In fact, 15% of tumours are estimated to be related to different infectious agents [119]. The first study analysing endometrial microbiome in patients with endometrial cancer revealed a compositional microbiome shift in the cancer cases, distinguishable from the benign cases [23]. The authors suggested that the presence of *Atopobium vaginae* and *Porphyromonas somerae* is associated with endometrial cancer [23]. Recently, the same group investigated endometrial cancer risks and they confirmed *Porphyromas somerae* presence as the most predictive microbial marker of endometrial cancer [24]. Meanwhile, another group studied women undergoing hysterectomy for endometrial tumours and detected *Acinetobacter*, *Pseudomonas*, *Cloacibacterium*, *Comamonadaceae*, and *Escherichia* as predominant taxa in the diseased endometrium [25].

## 4. Current Treatments in Clinical Practice

In the clinical setting, there is a high interest and demand to detect and to improve endometrial dysfunctions in order to treat uterine dysbiosis and to enhance infertility treatment outcomes. However, there is no unified protocol for assessing the endometrial microbial composition, neither for the treatment of uterine dysbiosis. 

Pioneering studies based on NGS approaches already developed commercially available tests for assessing endometrial microbiome: the EMMA test by iGenomix^®^ [120] and Endometrial Microbiome Test by Varinos Inc. [121]. The EMMA test is based on the previous study findings where *Lactobacillus* was dominating in the uterus, and that *Lactobacillus* dominance correlated with reduced miscarriage and implantation failures and thus improved pregnancy rates in women undergoing IVF [11]. EMMA test classifies endometrial samples into *Lactobacillus* dominant and non-*Lactobacillus* dominant profiles. Once the sample is classified as non-*Lactobacillus* dominant, adequate treatment including antibiotic, probiotics and prebiotics could be applied. In the same line, the Varinos test supports the *Lactobacillus* dominance in the uterus and categorises endometrial microbiome as *Lactobacillus* dominant and non-*Lactobacillus* dominant [9]. Then, choices for intervention are suggested, such as uterine lavage for microbial eradication, eradication treatment with antibiotics, and/or taking probiotics and prebiotics for improving the microbiota. Undoubtedly useful, these tests are based on mostly observational studies and include limited number of patients, resulting in rather limited evidence for testing and subsequent clinical decision-making in a clinical setting [122]. Clearly more research in the field is required starting with establishment of the core uterine microbiome before any treatment recommendations for ‘dysbiosis’ are offered for patients.

### 4.1. Antibiotics

The use of antibiotics is widely applied in gynaecology and obstetrics [123]. Antibiotics use ranges from diagnostic techniques (e.g., hysterosalpingography, sonography, hysteroscopy, and laparoscopy) to embryo transfer and up to diseases such as endometriosis and endometritis [123]. It is clear that the use of antibiotics produces fluctuation in microorganism communities within the female reproductive tract. 

In the infertility clinical setting, a broad-spectrum antibiotic therapy (i.e., amoxicillin or levofloxacin) was recently used to modulate the non-*Lactobacillus* dominant endometrial environment into *Lactobacillus* dominant, followed by the combination of antibiotics, prebiotics and/or probiotics, 53% (N = 17) of patients achieved *Lactobacillus* dominant state in the uterus while showing a higher but not statistically significant pregnancy rates compared to the non-*Lactobacillus* dominant group [9]. These results are found to be encouraging, and this treatment approach is gaining popularity in the clinics. 

CE is the gynaecological condition that is believed to benefit the most from the microbial modulation with antibiotics, as the bacterial infection in this condition is known [124]. Several studies showed that the use of antibiotics in CE has improved the reproductive outcomes. CE patients treated with antibiotics before embryo implantation had significantly better reproductive outcomes compared to those not treated with antibiotics [125], suggesting that the negative impact of CE on reproductive outcomes may be in part attributable to the presence of certain uterine bacteria. In line, McQueen et al. demonstrated that antibiotics treatment among the recurrent pregnancy loss patients with CE improved live birth rate from 7% to 56% after the treatment [126]. In a retrospective study conducted by Cicinell et al. the pregnancy and live birth rates were significantly higher in cured CE women treated with antibiotics when compared with the persistent CE group and even with the non-CE group [111]. Furthermore, Kitaya et al. reported that women with recurrent implantation failure (RIF) treated with antibiotics for CE had higher live birth rates compared with RIF patients with no CE [127]. In addition, the study conducted by Zhang et al. demonstrated that patients diagnosed with RIF and CE when treated with intrauterine antibiotic infusion therapy had significantly higher implantation and pregnancy rates when compared with RIF women without CE and with persistent CE groups [128]. Furthermore, intrauterine infusion of antibiotics after poor oral antibiotic therapy outcomes has also been applied in order to restore the physiological condition of the uterus in CE, and the three patients assessed achieved pregnancy after the treatment [129]. 

Women with endometriosis is another group of patients that could benefit from the uterine microbial modulation by antibiotics, as the hypothesis of ‘bacterial contamination’ in endometriosis was proposed [107]. Among patients who suffer endometriosis, the prophylactic antibiotic therapy was applied prior to oocyte retrieval with the aim to reduce infection; with the treatment the infection risk reached 0% [130]. In fact, prophylactic administration of antibiotics prior to oocyte retrieval in patients suffering from severe endometriosis is today commonly applied due to the risk of infection [123]. In animal models, mice with endometriosis were treated with broad-spectrum antibiotics and it resulted in a reduction of endometriotic lesions, supporting the possible role of microbes in endometriosis progression [131].

Undoubtedly effective in modulating uterine microbiome, best administration regimens and their effects on endometrial microbiome should be tested further in order to understand the impact on health and to avoid unnecessary antibiotic use.

### 4.2. Probiotics

Probiotics are defined as ‘live microorganisms that, when administered in adequate amounts, confer a health benefit in the host’, capturing the essence of probiotics, as microbial, viable and beneficial to the health [132]. Probiotic bacteria produce bioactive molecules that could act on the body and promote good health. These formulations might be a safe and effective alternative to antibiotics in restoring the imbalance of the uterine microbiota as seen in female urogenital disorders [133,134]. In the clinical setting, there is a high interest and demand to improve aberrant endometrial microbial environment in order to overcome dysbiosis and to enhance infertility treatment outcomes. 

Probiotic interventions were extensively investigated in relation to systemic inflammation [135]. The first probiotics were composed predominantly of members of *Lactobacillus* and *Bifidobacterium*, but they showed a lack of precision when targeting a specific biological function [136]. In the male reproductive field, the use of probiotics and synbiotics (formulations including probiotics and prebiotics) was related to improved semen quality [137,138], and in females the protective role of probiotics in relation to vaginal infections was observed [70,139]. A recent study aimed to select potential probiotic strains from vaginal samples to improve the health of the female reproductive tract, and selected the *Lactobacillus rhamnosus* BPL005 strain for its capacity to reduce pH and produce short chain fatty acids, protective effects against pathogen colonisation and lactate production [140]. 

Women suffering from endometriosis are shown to benefit from the administration of oral *Lactobacillus* by reduction in pain [141,142]. In the mouse model, the beneficial effect of *Lactobacillus* on endometriosis by increasing interleukin-12 levels and the activity of natural killer cells was suggested, and the administration of probiotics resulted in a reduction of endometriotic lesions [143,144,145]. 

Female genital microbiota modulation could also be used to fight or protect against infection. Recent research has demonstrated how probiotic lactobacilli (*Lactobacillus reuteri* RC-14 and *Lactobacillus rhamnosus* GR-1) can improve endometrial epithelial cells barrier function in response to human immunodeficiency virus-1 (HIV-1) [146]. It was also detected that bacterial strains could modulate the immune profile suggesting that the microbiota of the female reproductive tract is an important factor in the acquisition of resistance to the virus [146]. 

The field of probiotics in modulating uterine microenvironment is very promising, nevertheless we have to bear in mind that probiotics are not medical drugs and they often lack detailed product information, and only a few probiotics were investigated thoroughly in clinical trials [122]. 

### 4.3. Prebiotics

Prebiotics target microbes living in/on human with the result of improving health, providing non-viable substrates that serve as nutrients for beneficial microorganisms harboured by the host [147]. In an attempt to modify endometrial microbiome lactoferrin a prebiotic agent with favourable prebiotic activity was administered orally during and after treatment with antibiotics among women undergoing infertility treatment [9]. Among non-*Lactobacillus* dominant patients treated with lactoferrin for three month after the antibiotics therapy, 67% (6/9) of them reached *Lactobacillus* dominance in the endometrium [9]. Additionally, lactoferrin administration has demonstrated effective result against bacterial vaginosis, leading to pregnancy and full-term birth in women with a previous medical history of preterm birth [148]. 

Prebiotics use in the human reproductive field is in its infancy and it is for future studies to unravel its usefulness in modifying microbial niches in female reproductive tract.

### 4.4. Microbial Transplants

Faecal microbiota transfer (FMT) is increasingly being used for various indications; however, clear evidence for the efficacy of FMT currently exists only for recurrent *Clostridioides difficile* infection [149]. Over 100 clinical trials using FMT for different conditions are currently ongoing (ClinicalTrials.gov). The problem arises from the fact that FMT is not only being used in clinical trials, but also applied on individual patients with methods that are not publicly documented [149]. In this scenario, the required screening of FMT donors is not always performed in a standardised way, which can cause different side effects and complications among patients. In fact, the Food and Drug Administration (FDA, USA) has communicated a warning regarding the risk of severe bacterial infection after FMT [149]. The hypothesis for the uterine bacterial transmission route originating from the gut exists [30,150], thus FMT was proposed as a promising (future) tool for treatment female reproductive tract diseases [151]. It was shown on broad-spectrum antibiotics-treated endometriosis mice that the FMT from mice with endometriosis resumed the growth of endometriotic lesions suggesting that the gut microbiota could promote endometriosis progression [131]. In humans, differences in gut microbial composition between healthy women and women with endometriosis [152] and polycystic ovary syndrome were reported [153,154,155,156,157,158].

A new area of microbial transplants is arising—vaginal microbiota transplants (VMT), which is opening new frontiers for reproductive health [159]. VMT involves the transfer of cervicovaginal fluid from a ‘healthy’ donor to a patient who aims to restore the most beneficial microenvironment. A pioneering study has tested the use of VMT from healthy donors as therapeutic alternative for patients suffering from symptomatic, intractable and recurrent vaginosis, and reported positive treatment outcomes [160]. Since the uterine colonisation of microorganisms by vaginal-cervical ascension is known [35,36], VMT could open up a future way for managing endometrial dysbiosis. In addition, what about uterine microbiota transfer (UMT)? To conclude, microbial transplants are highly promising ways for modifying uterine microbiota, nevertheless thorough research and testing in randomised, placebo-controlled trials is warranted.

## 5. Challenges in Developing Targeted Modulators for Uterine Microbiota

The uterus represents an ideal organ for drug administration, possessing advantages such as the possibility of bypass first-pass metabolism, high permeability for low molecular weight drugs, considerable surface area for absorption, and rich blood supply [161]. However, the effectiveness of the site would depend on intrinsic factors that include pH, temperature, uterine fluid composition, viscosity, enzymatic metabolism, clearance, and others, together with the hormonal fluctuations throughout the menstrual cycle. For instance, uterine pH was shown to change along the menstrual cycle, oscillating between 6.4 and 7 [162] and this could modify the drug-release system since the vast majority of drugs possess an ionisable group (mostly weak bases) [163]. Other biophysical parameters such as oxygen tension (pO_2_) and temperature are also factors with probable influence on the modulators. pO_2_ is shown to have cyclical variation and minute-to-minute oscillations within human uterus [164]. Also, the temperature variation is cyclical by day and month, increasing in the luteal phase and is influenced by hormones, density of uterine vascular beds and effectiveness of local heat exchange [164]. It is clear that many factors could influence the drug-release system inside the uterus, and a detailed study of these properties are required in order to develop effective targeted microbiota modulators with the exact dose required, necessary time and right place.

Regardless of the different intrinsic factors, probably one of the most important challenges that arises in modulating the endometrial microbiome is the fact that it is a low microbial biomass niche [26,165]. It is clear that analyses of low-biomass microbial sites are sensitive to contamination (especially from lower genital tract) and data misinterpretation. Thus, researchers face hurdles when describing the baseline microbial communities in endometrium and require well-designed and well-controlled experiments in order to avoid and adjust for the risk of contamination [26,91]. This makes especially necessary to set up standardised detection and description methods for analysing the uterine microbial composition, as well as strategies to re-establish/maintain these microbial populations. The challenge of assessing the true microbial composition in endometrial eu- and dysbiosis, the adequate dosing and evaluating these effects remains for future studies.

## 6. Conclusions and Future Directions

The conventional approach to target bacterial dysbiosis has been and continues to be the use of antibiotics, which were shown to be both essential and effective for treating infections usually resulting from pathogen proliferation. Nevertheless, the antibiotic drugs lead to unintended off-target effects on microbial community structure which frequently causes adverse effects, making them less appealing as precise therapy to target microbiome [136]. In many cases, its administration aggravates the underlying dysbiosis in long term and may promote resistance [140]. For instance, prescription of antibiotics in a prophylactical way has not resulted in improved pregnancy outcome not even in the cases of high risk of preterm birth on the presence of pathogens [166].

New therapies, such as pro- and prebiotic administration, and microbiota transplants are gaining popularity for improving and maintaining the optimal composition of the microbiota (Figure 3). These approaches attempt to modulate the microbial communities in a way that is beneficial to human health. However, several important questions related to these new clinical strategies remain unresolved, such as indication for prescription, comparative efficacy of monostrain and multistrain probiotics, choice of excipients, and methods of administration and delivery [134]. Furthermore, procedures to develop reliable and reproducible microbiome-based therapeutic approaches represent a challenge [136]. Currently, the pharmacological mechanisms of probiotics and prebiotics’ action are poorly understood, providing not enough evidence to support the use of probiotics for medicinal purposes [134]. Of those commercially available probiotics, it is especially difficult to prove their clinical efficacy as they are based on studies on a small sample size, and the heterogeneity in strains of bacteria used, duration of treatment and the lifestyle of patients, which can also influence the effects of probiotic supplementation [167]. 

Regarding the human uterus, there are several studies where the effect of probiotic supplementation on the endometrial microbiota was studied, although mainly in combination with antibiotic treatment [9,140,141,142,145]. The use of alternative modulators for uterine microbes is a highly demanded and relevant area of investigation with direct clinical application. Nevertheless, before any treatment strategies could be offered, the core uterine microbial composition needs to be established. In fact, there is an active debate ongoing whether uterus harbours a unique microbiota or not [25]. If the microbiome is certainly present in the uterine environment in the absence of pathologic infection, current data support that it is of low abundance [26,168], and several technical challenges in studies of low-biomass samples exist and make difficult to distinguish microorganisms that are truly present in small quantities from those arisen from contamination [26,91,92]. Thus, the appropriate uterine microenvironment for non-pathological conditions has yet to be established and applying methods for targeted modification of microbial communities is premature. 

It is clear that in the case of infection caused by a pathogen, antibiotic treatment is required. However, in the case of prophylactics or suggested dysbiosis based on the molecular detecting methods (endometrial microbiome tests), today is too early to intervene and offer treatment recommendations for patients. In fact, no clinical recommendations are today available for diagnosis of ‘abnormal/unfavourable uterine microbiota’ [122]. To sum up, modulation of uterine microbiota for restoring and maintain microbial composition is a promising field of research and application with high clinical relevance, but we are not there yet and hopefully soon this promising future will be present.

## Figures and Tables

**Figure 1 biomolecules-10-00593-f001:**
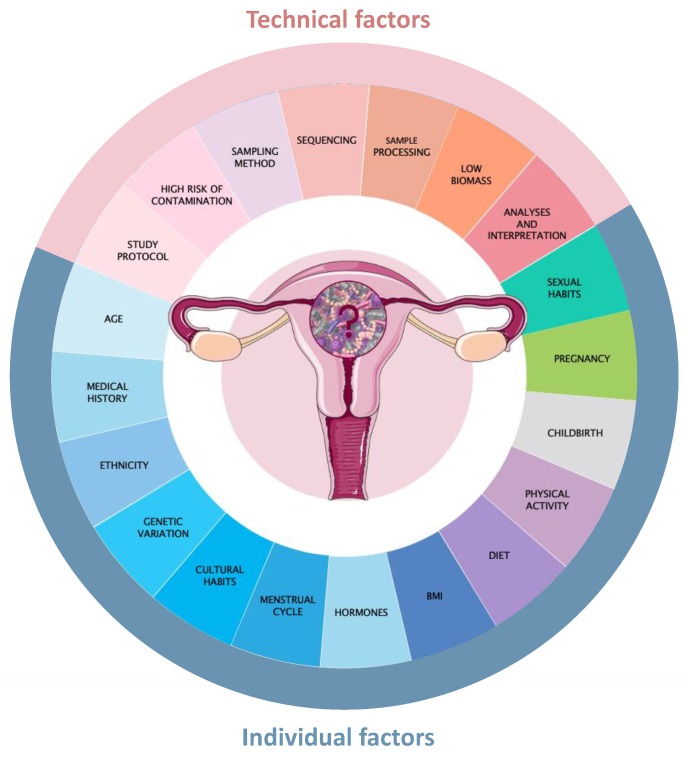
Dynamics and factors that could influence the endometrial microbiome and its analysis, including technical factors in the experimental design and individual/participant/group factors that could fluctuate the microbial composition in the uterus. Age: microbiome changes along lifespan [48], and the diversity and composition decline in the elderly [49,50]. Medical history: genital and extragenital diseases [19,22,23,51], including history of sexually transmitted infections can influence microbiome analysis. Ethnicity and different geographic regions: the ethnic origin of individuals seems to be an important factor to consider in microbiome research [52,53,54,55], as African American and Hispanic women show a trend to be more colonised with a non-*Lactobacillus* species in the upper genital tract than Caucasian women [34]. Genetic variation: host genetics can affect microbiome composition and maintenance [56,57]. Cultural habits: these habits showed leading to microbial variations [58]. Menstrual cycle and hormones: fluctuations in circulating oestrogen and progesterone levels might influence endometrial microbiome [19,57,59,60]. BMI: body mass index, especially obesity has been associated with altered intestinal microbiome composition [61,62]. Diet: it was shown that the gut microbiome not just metabolises ingested food but is itself shaped by the mode and type of food consumption [63,64]. Physical activity: exercising has shown to impact microbial composition [65,66]. Childbirth and pregnancy [67,68,69,70]. Sexual habits: the concept of seminovaginal microbiome, i.e., the partners share their microbial communities, has gained support [29]. Spermicidal agents could disrupt endometrial microbiota [71]; also sexual debut and activity can influence genital microbiome [72,73]. Study protocol: different protocols (collection technique, sample storage, extraction method, choice of primers) can lead to different results [74,75,76,77,78,79,80]. High risk of contamination: laboratory and reagent contamination can critically impact sequence-based microbiome analyses [81,82,83,84]; negative controls consisting of blanks are recommended [81,82,85,86]. Sampling method: an important point to consider consists in avoiding the surface of the vagina and the walls of cervical canal when taking the endometrial sample [87,88]. Sequencing: different sequencing platforms and sequencing filtering and processing can result in different outcomes [46,89,90]. Sample processing: amplicon sequencing has shown to cause great variance [75,90]. Low-biomass: high risk of misinterpretation of the results [26,91,92]. Analyses and interpretation: statistical analyses and interpretation can induce bias [84,90,93].

**Figure 2 biomolecules-10-00593-f002:**
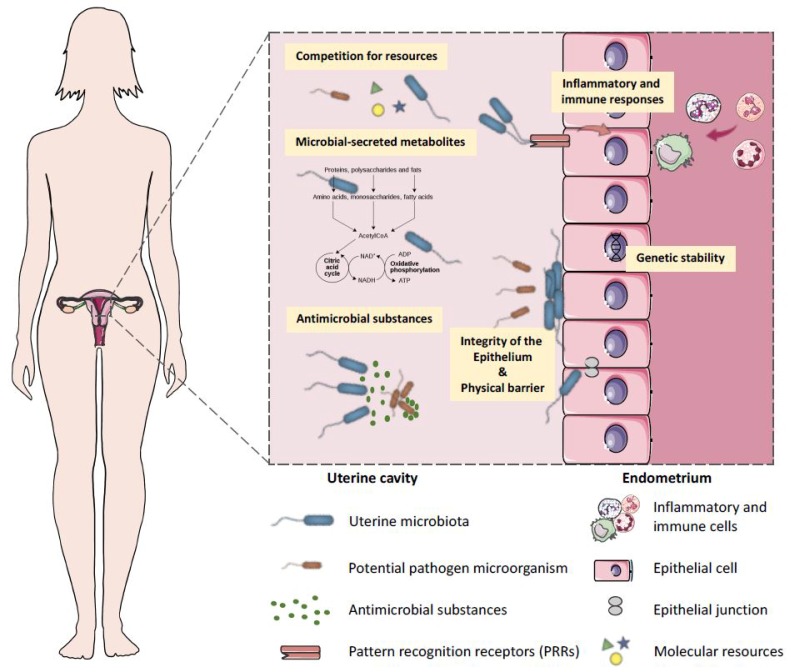
Endometrial microbiota interplay in the uterus. Uterine microbes could impact the genomic stability of uterine epithelia through modulation of transcription factors and other genomic and epigenetic alterations; also, microbial-secreted metabolites may support the growth of specific species and suppress the growth of other bacteria; and the consumption of a limited resource can starve the pathogenic invaders. In short, the endometrium is an immunologically suited niche for microbes: the endometrial immune system needs to be well adapted to withstand the continuous threat caused by microbial colonisation of the large endometrial mucosal surface, separated from host tissue by only a single layer of epithelial cells. Thus, tissue invasion of microbes must be limited in order to prevent potentially harmful inflammation or imbalance in the symbiotic relationship. Endometrial microbial homeostasis is probably regulated in three different ways: (1) a single layer of columnar epithelial cells forming a strong barrier through tight junctions anatomically limiting exposure of resident bacteria to the systemic immune system; (2) immune mediators such as infection-controlling molecules (antimicrobial peptides, AMPs) that are present in the endometrial mucosal surface and the endometrial fluid [99] and could restrict direct contact between epithelia and microbes; and (3) a rapid detection (epithelial cells express pattern recognition receptors (PRRs) that recognise and act to pathogens) and killing of bacteria upon a barrier breach by the endometrial lymphocytes that are present throughout all stages of the menstrual cycle [100,101].

**Figure 3 biomolecules-10-00593-f003:**
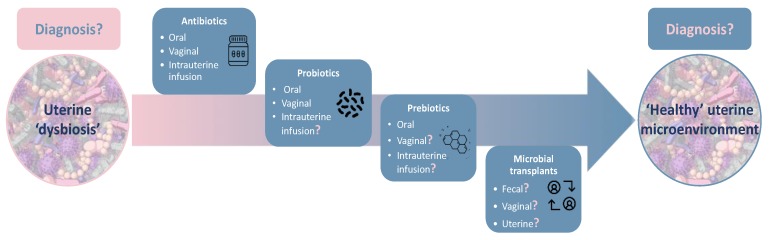
Current and future strategies for modifying uterine microbial composition. Intrauterine drug delivery represents an attractive alternative to achieve local and systemic effects due to the high contact surface exposed, the mucoadhesion of the epithelium, and the high absorption of drugs into the bloodstream. Several strategies for modifying endometrial microbial composition are being applied, nevertheless the core microbial composition is not established. The standard protocols for detecting uterine microbes and treatment protocols of dysbiosis are yet to be established.

**Table 1 biomolecules-10-00593-t001:** Predominant taxa in endometrial microbiome for different gynaecological disorders revealed by next-generation sequencing studies.

Gynaecological Condition	Predominant Taxa
Healthy	*Acinetobacter* [102,103], *Bacillus* [103], *Barnesiella* [23], *Bifidobacterium* [11], *Blautia* [23], *Corynebacterium* [102], *Desulfosporosinus* [16], *Enterobacter* [16], *Escherichia* [102,103], *Fusobacterium* [22], *Gardnerella* [11], *Jonquetella* [22], *Lactobacillus* [8,11,16,19], *Parabacteroides* [23], *Prevotella* [11,22], *Propionibacterium* [102], *Pseudomonas* [16], *Ralstonia* [16], *Shigella* [23], *Staphylococcus* [23,102], *Streptococcus* [11,102]
Infertility	*Atopobium* [6,8,9,10], *Bacteroides* [12], *Betaproteobacteria* [12], *Bifidobacterium* [5,8,9,10,11,13,104], *Burkholderia* [7], *Chitinophagaceae* [12], *Corynebacterium* [104], *Escherichia/Shigella* [10,12], *Flavobacterium* [4], *Gardnerella* [5,6,7,8,9,10,11,13], *Lactobacillus* [4,5,6,7,8,9,10,11,13,104], *Megasphaera* [9,10], *Pelomonas* [12], *Prevotella* [8,9,10,11,13], *Pseudoalteromonas* [5], *Rhodanobacter* [5], *Sneathia* [8,9], *Staphylococcus* [8,9,10,104], *Streptococcus* [6,7,8,9,11,104]
Endometriosis	*Acinetobacter* [20], *Barnesiella* [18], *Comamonadaceae* [20], *Enterobacteriaceae* [19], *Flavobacterium* [18], *Gardnerella* [17], *Lactobacillus* [17,18,20], *Moraxellaceae* [19], *Prevotella* [17], *Pseudomonas* [18,20], *Sphingobium* [20], *Staphylococaceae* [19], *Streptococcaceae* [17,19], *Vagococcus* [20]
Chronic endometritis	*Alteromonas* [16], *Anaerococcus* [15], *Atopobium* [15], *Bifidobacterium* [14,15,16], *Dialister* [15], *Gardnerella* [14,15,16], *Lactobacillus* [14,16], *Magasphaera* [14], *Parvimonas* [14], *Prevotella* [14,15], *Propionibacterium* [14], *Streptococcus* [14,16], *Veillonella* [14]
Endometrial polyps	*Alteromonas* [16], *Bifidobacterium* [16], *Euryarchaeota* (Archaea) [16], *Gardnerella* [16], *Lactobacillus* [16], *Streptococcus* [16]
Dysfunctional menstrual bleeding	*Gardnerella* [22], *Lactobacillus* [22], *Prevotella* [22], *Sneathia* [22], *Veilonella* [22]
Endometrial cancer	*Acinetobacter* [25], *Anaerostipes* [23], *Anaerotruncus* [23], *Arthrospira* [23], *Atopobium* [23], *Bacteroides* [23], *Cloacibacterium* [25], *Comamonadaceae* [25], *Dialister* [23], *Escherichia* [25], *Peptoniphilus* [23], *Porphyromonas* [23,24], *Pseudomonas* [25], *Ruminococcus* [23], *Treponema* [23]

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
