# Peer review of "New Opportunities for Endometrial Health by Modifying Uterine Microbial Composition: Present or Future?"

_biomolecules, 2020, doi:10.3390/biom10040593_

Round 1
Reviewer 1 Report
I found the article by Molina et al. a well organized and complete review of the current studies on endometrial microbiomes. In particular the article, focusing on the specific topic of possible therapeutic treatments, finds its place among the numerous available reviews in this field.
However, I have a major concern that is necessary to address to make the article acceptable for publication. It is related to the English. Even if I’m not an English native, I believe that there is room enough in the manuscript for improvements. Some sentences are too long, some are repetitive and also some grammar mistakes are present in the manuscript. I give some examples below with some suggestions of modifications (that you can further improve), but please don’t consider this list as exhaustive.
- Line 29-30: the sentence “Current knowledge suggests that the uterus harbours its own microbiota, where the microbes could influence the uterine functions in health and disease;” would be more correct if changed in: Current knowledge suggests that the uterus harbours its own microbiota, which could influence the uterine functions in health and disease;
- Line 47-48: for what it is reported below in this manuscript, I would suggest to write “unique uterine microbial compositions have been detected.”
- Lines 48-51: the sentence is long and repetitive. A possible alternative could be: As less than 1% of microbes can grow and form colonies on agar plates [2], analysis of the genomes of the microorganisms, i.e. microbiome, can provide information on genomic diversity [3], opening a new research field in reproductive medicine.
- Line 58-61: I would consider: Furthermore, the uterine microbial transmission is not clearly established. Different routes have been proposed including ascension of bacteria through the cervix, retrograde spread through fallopian tubes, haematogenous spread of oral and/or gut bacteria, gynaecological procedures (e.g. assisted reproductive technology-related procedures; insertion/removal of the 61 intrauterine devices), sexual habits, and/or sperm [26–30].
- Line 73-74: change into: Uterine sampling requires invasive methods,
- Line 78-80: the sentence needs to be rephrased.
- Lines 91-93: I would change into: Dynamics and factors that could influence the endometrial microbiome composition and its analysis, including technical factors in the experimental design and individual/participant/group factors.
- Lines 121- 124: Please make two distinct sentences.
- Lines 126-127: Maybe better: Endometrium seems to be an immunologically suited niche for microbiota, which could act in modulating inflammatory and immune responses.
- Line 128: exhibits
- Line 130: Are you meaning “provided by the fact”?
- Line 157: which are present.
- Lines 158-161: Since this is not properly a “barrier” abut an activity, I would suggest to modify the introductory sentence (lines 153-154) in something like this: Endometrial microbial homeostasis is probably regulated in three different ways:…
- Lines 181-184: The sentence should be rephrased
- Line 254: I would delete “and uterine environment”
- Line 257: approaches instead of sequencing
- Line 260: who/what does "their" refer to?
- Line 307: “prophylactic antibiotic therapy” instead of “prophylactic antibiotics”
- Line 330-334: I would make two distinct sentences.
- Line 336: with instead of by
- Line 357: results
- Line 401: it is influenced
- Line 411: their administration
- Line 422: represent
- Lines 424-427: rephrase the first part of the sentence.
Other points:
- Lines 78-80: Authors should also mention NGS methods useful for the detection of active bacteria, such as rRNA metabarcoding.
- Lines 102-106: gut microbiome changes related to different conditions (from BMI to physical activity) can influence the endometrial microbiomes. But why only these changes? Please clarify.
- Line 136: I believe that the sentence about the effect of microbiomes on the genomic stability of uterine epithelia is important enough to deserve some specific description.
Author Response
Reviewer 1 Comments to the Author
I found the article by Molina et al. a well organized and complete review of the current studies on endometrial microbiomes. In particular the article, focusing on the specific topic of possible therapeutic treatments, finds its place among the numerous available reviews in this field.
However, I have a major concern that is necessary to address to make the article acceptable for publication. It is related to the English. Even if I’m not an English native, I believe that there is room enough in the manuscript for improvements. Some sentences are too long, some are repetitive and also some grammar mistakes are present in the manuscript. I give some examples below with some suggestions of modifications (that you can further improve), but please don’t consider this list as exhaustive.
Line 29-30: the sentence “Current knowledge suggests that the uterus harbours its own microbiota, where the microbes could influence the uterine functions in health and disease;” would be more correct if changed in: Current knowledge suggests that the uterus harbours its own microbiota, which could influence the uterine functions in health and disease;
Line 47-48: for what it is reported below in this manuscript, I would suggest to write “unique uterine microbial compositions have been detected.”
Lines 48-51: the sentence is long and repetitive. A possible alternative could be: As less than 1% of microbes can grow and form colonies on agar plates [2], analysis of the genomes of the microorganisms, i.e. microbiome, can provide information on genomic diversity [3], opening a new research field in reproductive medicine.
Line 58-61: I would consider: Furthermore, the uterine microbial transmission is not clearly established. Different routes have been proposed including ascension of bacteria through the cervix, retrograde spread through fallopian tubes, haematogenous spread of oral and/or gut bacteria, gynaecological procedures (e.g. assisted reproductive technology-related procedures; insertion/removal of the 61 intrauterine devices), sexual habits, and/or sperm [26–30].
Line 73-74: change into: Uterine sampling requires invasive methods,
Line 78-80: the sentence needs to be rephrased.
Lines 91-93: I would change into: Dynamics and factors that could influence the endometrial microbiome composition and its analysis, including technical factors in the experimental design and individual/participant/group factors.
Lines 121- 124: Please make two distinct sentences.
Lines 126-127: Maybe better: Endometrium seems to be an immunologically suited niche for microbiota, which could act in modulating inflammatory and immune responses.
Line 128: exhibits
Line 130: Are you meaning “provided by the fact”?
Line 157: which are present.
Lines 158-161: Since this is not properly a “barrier” abut an activity, I would suggest to modify the introductory sentence (lines 153-154) in something like this: Endometrial microbial homeostasis is probably regulated in three different ways:…
Lines 181-184: The sentence should be rephrased
Line 254: I would delete “and uterine environment”
Line 257: approaches instead of sequencing
Line 260: who/what does "their" refer to?
Line 307: “prophylactic antibiotic therapy” instead of “prophylactic antibiotics”
Line 330-334: I would make two distinct sentences.
Line 336: with instead of by
Line 357: results
Line 401: it is influenced
Line 411: their administration
Line 422: represent
Lines 424-427: rephrase the first part of the sentence.
Authors’ response
We thank the Reviewer for the constructive comments and positive feedback. We believe that our manuscript has improved substantially. Regarding the English language, we have used the service of language revision by a native speaker, and do hope that it now meets the standards of adequate and good language (changes highlighted in the manuscript). Additionally, we have revised the comments one-by-one and have integrated the changes whenever appropriate.
Other points:
Lines 78-80: Authors should also mention NGS methods useful for the detection of active bacteria, such as rRNA metabarcoding.
Authors’ response
Thank you for the good comment. We have added a sentence to the manuscript (lines 84-86): ‘Furthermore, other NGS methods are being applied in order to detect active microorganisms, such as metatranscriptomics which analyses microbial RNA transcripts, identifying thereby potential functionally active microorganisms [46,47].’
Lines 102-106: gut microbiome changes related to different conditions (from BMI to physical activity) can influence the endometrial microbiomes. But why only these changes? Please clarify
Authors’ response
Regarding the legend of Figure 1, we aimed to highlight different factors that could influence the endometrial microbiome. Based on the current knowledge, however, many of the data (especially regarding BMI, diet and physical activity) are based on the studies in gut and today there are no studies published on female reproductive tract. The authors know that these studies are on its way and the aim here was just to highlight the broad spectrum of factors having role in influencing microbial composition.
Line 136: I believe that the sentence about the effect of microbiomes on the genomic stability of uterine epithelia is important enough to deserve some specific description.
Authors’ response
Thank you. We have added description regarding genomic stability to the sentence (lines141-146): ‘Also pathophysiological effects of uterine microbes on the endometrial epithelium have been proposed: 1) the genomic stability of uterine epithelia could be impacted via modulation of transcription factors and other genomic and epigenetic alterations; 2) the integrity of the epithelial barrier could be impaired; and 3) the microbial-secreted metabolites and the inflammation triggered by TLR activation could lead to suppression and/or overgrowth of specific bacteria [30].’ Additionally, Figure 2 legend gives further explanation to genomic stability (see page 5).
Reviewer 2 Report
The authors present a well referenced review of the current literature in the field of endometrial microbiome and reproductive health outcomes.
The review is structured in a logical manner leading the reader from the central dogma of a sterile uterus to the modern day appreciation of an endogenous microbiota throughout the female reproductive tract. I would suggest that more emphasis be placed on the low-biomass uterine microbiome, and this the challenges in moderating imbalance at this site using microbial transplantation methods.
figures should be reviewed for typographical errors.
Author Response
Reviewer 2 Comments to the Author
The authors present a well referenced review of the current literature in the field of endometrial microbiome and reproductive health outcomes.
The review is structured in a logical manner leading the reader from the central dogma of a sterile uterus to the modern day appreciation of an endogenous microbiota throughout the female reproductive tract. I would suggest that more emphasis be placed on the low-biomass uterine microbiome, and this the challenges in moderating imbalance at this site using microbial transplantation methods.
Figures should be reviewed for typographical errors.
Authors’ response
We thank the Reviewer for the positive feedback and good comments. The typographical errors have been corrected, and additionally the manuscript has been revised by a native speaker. We agree that the low-biomass uterine microbiome is an important issue and we have elaborated on the topic (please see lines 437-488): ‘Regardless of the different intrinsic factors, probably one of the most important challenges that arise in modulating the endometrial microbiome is the fact that it is a low microbial biomass niche [26,163]. It is clear that analyses of low-biomass microbial sites are sensitive to contamination (especially from lower genital tract) and data misinterpretation. Thus, researchers face hurdles when describing the baseline microbial communities in endometrium and require well-designed and well-controlled experiments in order to avoid and adjust for the risk of contamination [26,91]. This makes especially necessary to set up standardised detection and description methods for analysing the uterine microbial composition, as well as strategies to re-establish/maintain these microbial populations. The challenge of assessing the true microbial composition in endometrial eu- and dysbiosis, the adequate dosing and evaluating these effects remains for future studies.’
Round 2
Reviewer 1 Report
Most of the changes required have been made. Manuscript acceptable for publication.